# Synthesis and Evaluation of Novel Ellipticines and Derivatives as Inhibitors of *Phytophthora infestans*

**DOI:** 10.3390/pathogens9070558

**Published:** 2020-07-10

**Authors:** Mary L. McKee, Limian Zheng, Elaine C. O’Sullivan, Roberta A. Kehoe, Barbara M. Doyle Prestwich, John J. Mackrill, Florence O. McCarthy

**Affiliations:** 1School of Chemistry and Analytical and Biological Chemistry Research Facility, University College Cork, Western Road, T12 K8AF Cork, Ireland; m.mckee@umail.ucc.ie (M.L.M.); elosullivan@hotmail.com (E.C.O.); 118225550@umail.ucc.ie (R.A.K.); 2Department of Physiology, School of Medicine, University College Cork, Western Road, T12 K8AF Cork, Ireland; 107139861@umail.ucc.ie; 3School of Biological, Earth and Environmental Sciences, University College Cork, Western Road, T12 K8AF Cork, Ireland; b.doyle@ucc.ie

**Keywords:** *Phytophthora infestans*, ellipticinium salts, isoellipticines, mycelial growth, oomycete, zoosporogenesis

## Abstract

The pathogen *Phytophthora infestans* is responsible for worldwide catastrophic crop damage and discovery of new inhibitors of this organism is of paramount agricultural and industrial importance. Current strategies for crop treatment are inadequate with limitations of efficacy and market alternatives. Ellipticines have recently been reported to have fungicidal properties and have been assessed against *P. infestans* growth with promising results. We hereby report a probe of the ellipticine framework to investigate the alkyl subunit and screen a set ellipticines and derivatives to identify new lead compounds to act against *P. infestans*. A series of ellipticinium salt derivatives have been identified with exceptional growth inhibitory activity and apparent lack of toxicity towards a human cell-line surpassing the effect of known and marketed fungicides. This report identifies the potential of this natural product derivative as a novel fungicide.

## 1. Introduction

One of the most significant threats to global food production and agriculture is the pathogenic oomycete *Phytophthora infestans* (*P. infestans*) [1]. Famously known for the Irish potato famine in 1845-1849, the pathogen causes devastating outbreaks of late potato blight and continues to be a worldwide problem to this day [2,3]. The global impact of *P. infestans* is in excess of $6.2 billion per annum due to crop loss and the use of fungicides [4,5]. The necessity of fungicides to combat the spread of *P. infestans* has caused resistant strains to continually develop, and so the design of novel fungicides to target this pathogen would help address one of the fundamental challenges of the 21st century-food security [6].

The first ever pesticide to be adopted, the Bordeaux mixture, was used to combat *P. infestans* blight [7]. The emergence of this fungicidal treatment was imperative to the conservation of crops and averting hunger, though its use diminished through the emergence of less toxic and more targeted fungicides [8]. In the 1960s it was replaced by the dithiocarbamate class of broad spectrum fungicides due to their high activity and low production cost [9]. Mancozeb (**1**) and Propineb (**3**) were two of the most widely used dithiocarbamates however, both have been found to breakdown into toxic metabolites following degradation (Figure 1) [10,11,12,13]. 

Following dithiocarbamates, phenylamide fungicides were developed including Metalaxyl (**5**) (Figure 2); however, the first resistant plants were identified after seven years [14]. This finding has resulted in formulation in combination with other fungicides (e.g., Mancozeb, **1**) which results in the reduction of emerging resistant strains [15]. 

Approved by the EU in 2017, oxathiapiprolin (**6**) was developed as an anti-oomycete fungicide (Figure 2). It was designed and synthesised based on the structure of a piperidine-thiazole-carbonyl core [16]. Oxathiapiprolin (**6**) has been approved in many countries around the world and is more potent, allowing lower overall dosage [17]. However, similarly to other treatment regimens, oxathiapiprolin has already been found to be ineffective against emerging strains of *Phytophthora capsici* oomycetes that have resistance and hence the future of the current market in fungicides is uncertain [18]. 

In recent years, focus has turned towards the use of natural products as fungicides and although the first report of anti-microbial natural products was in 1676, development has been limited [19]. A prime avenue of interest for natural source fungicides is the essential oil, cinnamaldehyde (**7**, Figure 3) and we have recently published data to suggest that derivatives of cinnamaldehyde which modify the aldehyde functional group could prove important in future studies [20,21]. A separate outcome of this research identified aldehyde derivatives of the natural product ellipticine (**8**) as a potential lead compound [21]. 

Cinnamaldehyde is a natural product extracted from the bark of *Cinnamomum* plant genus. This conjugated aromatic compound has shown to inhibit the growth of bacteria, filamentous moulds and yeast [20]. The essential oil is generally recognised as safe by the United States Food and Drug Administration and has been used in foods and medicines as flavourings for many years [22]. Extensive research has also been carried out on the anti-bacterial properties of cinnamaldehyde, demonstrating strong activity against *Bacillus, Staphylococcus* and *Enterobacter* spp. and can therefore be used as a food preservative [23,24,25]. One of the key components of cinnamaldehyde is the aldehyde which is also seen in other active cinnamaldehydes (**9**–**13**, Figure 4) and we originally set out to explore this with a series of aromatic aldehydes [21,26]. 

A recent study by Watamoto describes a diverse set compounds which demonstrate inhibitory effects on fungal strains, amongst which some of the structural features of our targets align [27]. Two hit compounds from this screen (**14** and **15**, Figure 4) have strong fungicidal effects and inhibit the metabolic activity of *C. albicans*. Of these, Bay 11-7082 (**14**) possesses an α, β-unsaturated bond similar to cinnamaldehyde and may act through similar mechanism. Ellipticine (**15**) has known antimicrobial and especially anticancer properties: Testing at bacterial planktonic mode showed (**15**) had averaged a minimal inhibitory concentration (MIC) of 8.45 μM across all six *Candida* strains and hence is worthy of further investigation [28,29,30]. This was the basis for our investigation into ellipticines which identified that 9-formyl ellipticines have exceptional *P. infestans* growth inhibition characteristics. We have reported recently that ellipticines **15**–**19** (Figure 5) have with excellent *P. infestans* growth and zoosporogenesis inhibitory properties, but which are limited by potential toxicity against human cells [21]. Given the need for new fungicidal compounds to address the market need, we set out to probe the interaction by structural modification of ellipticine. 

The aims of this study are to synthesise 2-and 6-substituted 9-formylellipticine derivatives by expansion with alkyl substituents and evaluate these new compounds in assays of *P. infestans* mycelial growth and zoosporogenesis (the production of the motile zoospore stage). Interestingly, one of the fungicidal compounds identified by Watamoto (CV-3988) possesses a long alkyl chain and had fungicidal effects on Candida strains, but low cytotoxicity on human cells [27]. Ellipticine derivatives are accessed by specific synthetic alkyl modification at the 2- and 6-positions to probe the influence of solubility and the inclusion of isoellipticines to test structural effects (Figure 6) on growth inhibition and potential cytotoxicity. We report the identification of a series of potent ellipticinium salts as lead compounds for field studies. 

## 2. Results

Synthesis of novel ellipticine derivatives was undertaken and the products were assessed in a parallel approach, via the determination of inhibitory effects on the growth of *Phytophthora infestans* and cytotoxicity against human cells (XTT assay). 

### 2.1. Synthesis of Novel Ellipticine Derivatives

Ellipticines **15**–**19** have been reported recently with excellent *P. infestans* growth inhibitory properties and initially were used as the basis for substitution at the 2-position [21,31,32].

#### 2.1.1. Synthesis of Ellipticinium Salts 

Alkylation at the 2-position is commonly used to increase the aqueous solubility of ellipticines and derive new bioactivity as it significantly affects the electronics of the system in forming a charged salt [33,34,35]. It was envisaged that this modification would affect the *P. infestans* inhibitory properties and the influence of alkyl chain length and terminal group was to be assessed. Compounds **18** and **19** were chosen as leads given their identified *P. infestans* growth inhibitory properties and the potential for generation of new chemical entities.

Synthesis of compounds **20**–**29** proceeded in a facile manner with the relevant alkyl halide and the water soluble salts were isolated in good yield and purity (Scheme 1, Table 1). The only exception to this is isopropyl salt **22** and which is presumably due to steric hindrance of the incoming electrophile.

The choice of alkyl group was intended to probe long and short alkyl chains (to mimic the long chain seen in CV-3988) and those with a electrophilic functional group (nitrile) to test the influence on growth inhibition. This set of 2-alkyl substituted ellipticinium salts was readied for screening of *P. infestans* growth inhibition and cytotoxicity assessment.

#### 2.1.2. Synthesis of Isoellipticines and Isoellipticinium Salts

Given the known activity of ellipticines, it was of interest to probe the influence of the ring nitrogen position and hence the related isoellipticines were generated [34,36]. Synthesis of isoellipticines was undertaken via the route of Gribble et al. in an analogous manner to that of ellipticine **15** and generated isoellipticine (**30**) (Figure 7, Appendix A) [32]. This was then functionalised in one of two ways in order to probe the interaction: Isoellipticine (**30**) was alkylated at the 10-position and the relevant 7-formyl substitution completed (compound **31** and **32**) [37]; 7-formyl isoellipticine was alkylated at the 2-position with a methyl substituent (compound **33**) [34]. Compounds **30**–**32** allow for direct comparison with the ellipticines which had been identified previously (Figure 5) and the isoellipticine salt **33** provides comparison points with the new synthetic library from Section 2.1.1.

### 2.2. Screening of Derivatives of Ellipticine and Ellipticinium Salts against P. infestans

Nineteen synthetic ellipticine derivatives were assessed in a parallel approach, via the determination of inhibitory effects on the mycelial growth, zoosporogenesis and zoospore motility in *Phytophthora infestans* and cytotoxicity against a mammalian cell-line, human embryonic kidney-293T (HEK-293T) and all data reported is the mean of three individual experiments [38,39,40,41,42]. 

Initially, the compounds were tested in a *P. infestans* bioassay by individually mixing with agar and setting into separate petri-dishes at 25 μM concentration. A plug of *P. infestans* was then placed on top of the agar and mycelium growth monitored periodically. The growth restriction of the pathogenic mycelium outward from the plug shows the inhibitory effects of each compound on *P. infestans*. 

#### 2.2.1. *Phytophthora infestans* Mycelium Growth Assay

Compounds were evaluated for relative inhibitory effects on *P. infestans* at a concentration 25 μM for mycelium growth after five, nine and thirteen days. Of note for this assay, Mancozeb (**1**), the commercial pesticide, is at a test concentration of 100 μM and cinnamaldehyde (**7**) was also examined for comparison at a concentration of 1 mM [38]. 

##### Day 5 *P. infestans* Growth Inhibition by Ellipticines and Derivatives 

Cinnamaldehyde (**7**), 9-formyl-6-methylellipticine **17** and short chain salts (**20**–**22**) exhibit the best inhibition of *P. infestans* with 0% growth for all five compounds (Figure 8). Demonstrating the least inhibitory effects are isoellipticine **30** and long chain salts (**24**, **25** and **28**) at 138%, 131%, 150% and 119% respectfully when compared to the control growth. There is a remarkable difference in the activity of differing chain lengths for the alkyl substituent at the 2-position (comparing **20**–**22** to **24**–**25** and **28**–**29**). The inactivity of isoellipticine **30** is potentially due to the insolubility of isoellipticine but interestingly the 10-methylisoellipticine **31** is associated with improved activity and mirrors the effect seen between the ellipticines (**15**) and **16**. This highlights the importance of the nature of the alkyl group attached to either nitrogen.

It is possible that inhibitory effects are related to the length of the *N*-2 alkyl chain with the methyl, ethyl and isopropyl groups demonstrating excellent inhibition relative to the long chain salts. Although Mancozeb (**1**) is four times the concentration of the remaining test compounds, it has a relatively low inhibition effect compared with seven other test compounds demonstrating more potent effects.

##### Day 9 *P. infestans* Growth Inhibition by Ellipticines and Derivatives

After nine days, it can be seen once again that cinnamaldehyde (**7**), **17**, **20**, and **21** demonstrate 100% inhibition of mycelium growth (Figure 9). There has been an increase of 19% in mycelium growth for **22** which infers that the isopropyl salt is either not as potent or as stable in this medium as the methyl or ethyl ellipticinium salt.

Compounds **30**, **24**, **25** and **28** still demonstrate the weakest inhibition rates of all the test compounds; however, the rate of mycelium growth has decreased in each case. Mancozeb (**1**) shows an increase of 13% mycelium growth since day five.

##### Day 13 *P. infestans* Growth Inhibition by Ellipticines and Derivatives

It is clear from Figure 10 that there is a high rate of mycelial growth for most compounds after thirteen days. However, there is still a consistent 0% growth for cinnamaldehyde (**7**), **20** and **21**. This points to definitive fungicidal activity for ellipticinium salts **20** and **21** and given the dosage difference to that of cinnmaldehyde shows significant promise. The growth rate has decreased for **24**, **25**, **28** and **30** to below that of the Rye B control. Mancozeb (**1**) has an increased rate of mycelium growth of 14% since day nine.

##### Concentration Dependence of Cinnamaldehyde, (7), and Compound 21 on P. infestans Mycelial Growth

In order to compare the efficacy of cinnamaldehyde, (**7**), with a lead compound identified in the current study, **21**, the RyeB mycelial growth assay was performed with six different concentrations of each compound (1, 2, 5, 10, 20, 50 and 100 μM for **21**; 10, 20, 50, 100, 200, 500 and 1000 μM for **7**). This permitted the construction of concentration-colony diameter relationships and estimation of half-maximal inhibitory concentrations from these by curve-fitting to a four-parameter logistic equation, Figure 11. This revealed that compound **21** is nearly 50 times more effective than cinnamaldehyde at inhibiting *P. infestans* mycelial growth.

#### 2.2.2. Effects on Zoosporogenesis

In addition to mycelium growth inhibition, the ellipticine compounds were assessed for their effects on zoosporogenesis, the production of motile zoospores. The number of zoospores growing in each sample were counted using a haemocytometer as per the method described (see Section 4). Mancozeb (**1**) had no significant inhibitory effect on the production of zoospores. In addition to cinnamaldehyde, eight compounds demonstrated complete inhibition of zoospore formation with the *N*-2 alkyl ellipticinium salts, **20** and **21** included (Figure 12). This further supports the argument that compounds **20**–**22** are the strongest anti-oomycete candidates from the current study. In addition, the relationship between inhibition of zoosporogenesis and reduction of mycelial growth is not simple, with some compounds completely ablating zoospore production, without abolishing hyphal growth, see Figure 10 and Figure 12. 

#### 2.2.3. Effects on Zoopore Motility

Zoospores are the motile, asexual life-cycle stage of oomycetes and represent one of the mechanisms by which infections propagate. In order to determine the effect of selected compounds on their motility, zoospores were collected from 13 day cultures of *P. infestans* on RyeB agar, and their movement recorded using brightfield video microscopy. As described in Section 4, analysis of the recordings revealed zoospore velocity (which is directly related to total distance travelled), Euclidean distance (length of a straight-line from origin to the furthest point travelled) and directionality (how straight the motility is, with a value of 1 being travel in completely straight line). These experiments revealed that only Mancozeb and compound **30** had any effects of zoospore motility, Figure 13. Both Mancozeb and **30** significantly decreased the average velocity of zoospores. Only Mancozeb influenced directionality, with zoospores travelling in straighter paths.

#### 2.2.4. Human Embryonic Kidney Cell XTT Assay

An XTT cytotoxicity assay was utilised to assess the toxicity of compounds against Human Embryonic Kidney 293T (HEK-293T) cells [41,42]. This is performed at a concentration of 25 μM to provide an indication as to which structures are the least toxic to humans. Cinnamaldehyde was used for comparison at a concentration of 1 mM; however, the active concentration of cinnamaldehyde was determined by Hu et al. to be 2 mM to completely inhibit mycelial growth [38]. Mancozeb **1**, a commercial pesticide, was also tested at 100 μM as this was found to be the active anti-oomycete concentration by Shin et al. [40]. 

##### Results from XTT Bioassay 

The parent compound ellipticine (**15**) has a low cell viability of 3% which confirms the known cytotoxicity of the natural product (Figure 14). Substitution at the *N*-6 position to, for example, 6-methylellipticine **16** lowers the toxicity to 37% which lowers further to 40% upon addition of a formyl group at the 9-position to form 9-formyl-6-methylellipticine **17**. The lowering of toxicity is an essential requirement for these compounds to progress to field trials and its importance is shown here. In addition to substitution playing a vital role in toxicity, the size of the substituted group also has an effect. In comparison to 9-formyl-6-methylellipticine **17** (cell viability of 40%), 9-formyl-6-ethylellipticine **18** and 9-formyl-6-isopropylellipticine **19** show cell viabilities of 14% and 82% respectfully. It is evident that the isopropyl substituent expresses the highest cell viability for the ellipticine compounds and validates the rationale for compounds **26**–**29**. 

The *N*-2 quaternary salts vary in potency significantly. Compound **25** has an 18-carbon chain with a moderate cell viability of 61%: however, reducing the chain to 14-carbons **24** results in significantly lower cell viability of 3%. Compounds **20**, **21** and **22** which have methyl, ethyl and isopropyl *N*-2 substitution are reported with moderate toxicities of 61%, 75% and 56% respectfully. It is evident that the addition of a short chain increased cell viability over both unsubstituted and long chain substituted, therefore, it is plausible that the *N*-2 substituent can impact cell viability. This is an important finding given the *P. infestans* growth inhibitory activity of **20**–**22**.

Ellipticine (**15**) and isoellipticine (**30**) have respective toxicities of 3% and 1%. The two isomers, 9-formyl-6-methylellipticine **17** and 7-formyl-10-methylisoellipticine **32** have toxicities of 40% and 22% respectfully. Comparison of ellipticine and isoellipticine also indicates that the position of the pyridine nitrogen on the D-ring plays a role in cell viability. The least toxic compounds in the series are **31** and **33** which both arise from isoellipticine but have unique substitution patterns.

## 3. Discussion

Relationships can be drawn from the data relating the compound structure to the growth inhibitory effects against *P. infestans*. Data from day five will be used for this section as it represents the initial growth rate of the mycelium. 

The parent compound ellipticine (**15**), allowed a high mycelium growth rate of 106% relative to the control. When this compound is methylated at the *N*-6 position to generate **16**, the growth rate drops to 21% which lowers to 0% when **16** is formylated at the 9-position to form **17**. It is postulated that protecting the *N*-6 position plays a vital role in lowering the growth rate of *P. infestans* and addition of an aldehyde group at the C-9 position results in complete inhibition of growth [21].

It appears that not only is substituting the *N*-6 site important, the alkyl group that is substituted also plays a role. 9-Formyl-6-methylellipticine **17** shows 0% growth of *P. infestans* whereas 9-formyl-6-ethylellipticine **18** has 69% growth and 9-formyl-6-isopropylellipticine **19** demonstrates 56% growth. Evidently, the size of the substituent at the *N*-6 position impacts the potency of ellipticine derivatives [21]. 

Quaternary salt formation at the *N*-2 position has also been shown to impact the potency of ellipticines [33,34]. Compound **25** is *N*-2 substituted with an 18-carbon chain is the least potent compound in the assay with a growth rate of 150% followed by the 14-carbon chain salt **24** with 131%. Compound **23** has a 6-carbon chain with a terminal nitrile exhibiting 102% mycelium growth. As mentioned before, each of the methyl, ethyl and isopropyl salts show a 0% growth of *P. infestans* highlighting the impact of the shorter chain salts on growth. In assessing the specific effect of the 6-substituent on the ellipticinium salts, there is a general trend for the 6-ethyl salts (**20**–**25**) being more potent than the 6-isopropyl (**26**–**29**).

Remarkably, the effects of the ellipticinium salts on *P. infestans* growth from one 25 μM dose last through 9 and beyond 13 days (and even as far as 35 days—data not included). This is particularly true of compounds **20** and **21** which still register 0% growth with limited effects also seen for **22**, **26**, **27**, **31** and **32** and all other compounds registering no remaining inhibitory effects which serves to reinforce the importance of short chain alkyl substituents for inhibition. The effects on zoosporogenesis across the panel are also highly instructive with **20**–**22** and **31**–**32** effectively abolishing the production of motile zoospores.

Ellipticine (**15**) and isoellipticine (**30**) parent compounds also demonstrate different effects on *P. infestans* (106% vs 138%). Comparing the related compounds from each series (**15** vs. **30**, **16** vs. **31**, **17** vs. **32** and **20** vs. **33**) it is clear that the position of the pyridine nitrogen on the D-ring plays a role with the ellipticines generally more potent than their isoellipticine counterparts though this relationship requires further development. 

Of key importance in the development of new agents for crop protection will be the differential between effective concentration for pathogenic growth inhibition and cytotoxicity to mammalian cells. To this end we can compare the data from Day 5 of mycelium growth and the XTT assay (Figure 15).

Compounds **20**, **21** and **22** possess similar mycelium growth properties, however, the three compounds do not have the same level of cytotoxicity. Although close in value, the XTT assay data shows that **22** has the highest toxicity of the three salts at 56% followed by **20** at 61% and the least toxic of the three **21** at 75%. The toxicity of incorporating the alkyl chain increases at the 14-carbon chain **24** to 3% cell viability and decreases at 18-carbon length chain **25** to 61%. It is plausible that toxicity is more sensitive to changes in compound structure based on this data. From the information obtained, it is apparent that **21** is the best candidate from this study as it has relatively low toxicity and excellent inhibitory effects against *P. infestans* with the highest inhibitory percentage throughout the study. In looking at the opposite effect, compounds (**15**) and (**30**), in addition to **24** and **28** have the least desirable profile in respect of safety concerns. The consistency between these compounds (ellipticine/isoellipticine and two C14 alkyl ellipticinium salts) is remarkable and could serve as future templates to focus on where cytotoxicity is necessary (e.g., proliferative diseases).

The comparison between cinnamaldehyde (**7**) and **21** (Figure 11) highlights the potency (>50 times) of the ellipticinium salt over the natural product and when added to the effects seen on zoosporogenesis should serve as a useful reference for future investigations. 

## 4. Materials and Methods 

### 4.1. General Procedures

Solvents were distilled prior to use by the following methods: Ethyl acetate was distilled from potassium carbonate; THF was freshly distilled from sodium and benzophenone and hexane was distilled prior to use. Organic phases were dried using anhydrous magnesium sulfate.

All commercial reagents were used without further purification unless otherwise stated. Alkyllithium reagents were titrated prior to use using the Gilman double titration procedure as follows: Two 100 mL conical flasks were prepared, the first one containing water (25 mL) and the second one containing dibromoethane (2 mL, stoppered with a SubaSeal under nitrogen with a provision for pressure release). The alkyllithium reagent (1 mL) was added via syringe to each flask [43,44]. The reaction with dibromoethane was more vigorous than that with water, and the flask was swirled during addition. Two drops of phenolphthalein were added to both flasks. The first flask was titrated against 0.1 M HCl until the purple colour disappeared. The SubaSeal was removed from the second flask and water (25 mL) was added, forming a biphasic mixture and was titrated as before. (Note: Constant stirring was maintained to ensure mixing of both phases during titration). Titre value 1 represents the total base (alkyllithium and inorganic base) while titre 2 represents the free base (inorganic base not resulting from the alkyllithium). Thus, the molarity of the alkyllithium reagent was calculated using the following equation: molarity of alkyllithium (M) = [(titre 1 – titre 2) × 0.1].

In reactions utilising alkyllithium and sodium hydride, all glassware was flame dried under nitrogen prior to use. All low temperature reactions were carried out in a three-necked round bottomed flask equipped with a low temperature thermometer and the internal temperatures are quoted. Syringes were used to transfer small volumes of alkyllithium reagents while cannulation from the reagent bottle into a pre-calibrated addition funnel was used for larger volumes (>15 mL). Low temperature reactions used the following cooling mixture: −100 °C, absolute ethanol and liquid nitrogen. 

^1^H (300 MHz) and ^13^C (75.5 MHz) NMR spectra were recorded on a Bruker AVANCE 300 NMR spectrometer. ^1^H (600 MHz) and ^13^C (150.9 MHz) NMR spectra were recorded on a Bruker AVANCE III 600 NMR spectrometer equipped with a Bruker Dual C/H cryoprobe or a Bruker Broadband Observe H&F cryoprobe. All spectra were recorded at 300 K (26.9 °C) in deuterated dimethylsulfoxide (DMSO-*d6*) using DMSO-*d6* as the reference peak or in deuterated chloroform (CDCl_3_) using trimethylsilane as an internal standard unless otherwise specified. Chemical shifts (δH and δC) are reported in parts per million (ppm) relative to the reference peak. Coupling constants (J) are expressed in Hertz (Hz). Splitting patterns in ^1^H spectra are designated as s (singlet), br s (broad singlet), d (doublet), t (triplet), br t (broad triplet), q (quartet), sept (septet), dd (doublet of doublets), ddd (doublet of doublet of doublets) and m (multiplet). Signal assignments were supported by COSY (correlation spectroscopy) or HMBC (Heteronuclear Multiple-Bond Correlation spectroscopy) experiments where necessary. ^13^C NMR spectra were assigned (aromatic C, CH, CH2, CH3) with the aid of DEPT (Distortionless Enhancement by Polarisation Transfer) experiments run in DEPT-90, DEPT-135 and DEPT-q modes. Specific assignments were made using HSQC (Heteronuclear Single Quantum Correlation) and HMBC (Heteronuclear Multiple Bond Correlation) experiments. All spectroscopic data for known compounds was in agreement with those previously reported unless otherwise stated. Samples used for comparison (stacked plots) were run at equal concentrations (6-8 mg per 0.65 mL solvent). Carbon analysis is given for compounds that are novel or where full analysis has not been published in the literature. 

Infrared spectra were recorded on a Bruker Tensor 37 FT-IR spectrophotometer interfaced with Opus version 7.2.139.1294 over a range of 400–4000 cm^−1^. An average of 16 scans was taken for each spectrum obtained with a resolution of 4 cm-1. Melting points were measured on a Uni-Melt Thomas Hoover capillary melting point apparatus and are uncorrected. Melting points or boiling points were not obtained for semi-solids or oils. 

Thin Layer Chromatography (TLC) was carried out on precoated silica gel plates (Merck 60 F254). Visualisation was achieved by UV light detection (254 nm or 366 nm), Wet flash chromatography was carried out using Kieselgel silica gel 60, 0.040–0.063 mm (Merck). 

Low resolution mass spectra were recorded on a Waters Quattro Micro triple quadrupole spectrometer (QAA1202) in electron spray ionisation mode (ESI) using acetonitrile:water (1:1) containing 0.1% Formic acid as eluent. High resolution mass spectrometry (HRMS) spectra were recorded on Waters Vion IMS (model no. SAA055K) in electron spray ionisation mode (ESI) using acetonitrile:water (1:1) containing 0.1% Formic acid as eluent.

### 4.2. 6-Ethyl-9-Formyl-2,5,11-Trimethyl-6H-Pyrido[4,3-b]Carbazol-2-Ium Iodide **20**

Iodomethane (0.034 mL, 0.078 g, 0.552 mmol) was added to a stirred suspension of 6-ethyl-5,11-dimethyl-6*H*-pyrido[4,3-*b*]carbazole-9-carbaldehyde **18** (0.152 g, 0.501 mmol) in anhydrous DMF (3.5 mL) and stirred at room temperature for 18 h. The reaction mixture was cooled on ice and cold diethyl ether (3 mL) was added. The resulting precipitate was collected by vacuum filtration and washed with hexane (3 mL) and diethyl ether (5 mL) to give the product as a yellow solid which was dried at 0.2 mbar (0.175 g, 78.8%). m.p. > 300 °C; v_max_/cm^−1^ (KBr): 3072, 3047, 2934, 1672, 1573, 1242, 836; δ_H_ (300 MHz, DMSO-*d*_6_): 1.46 [3H, t, 7.0, N(6)CH_2_CH_3_], 3.09 [3H, s, C(5)CH_3_], 3.31 [3H, s C(11)CH_3_], 4.49 [3H, s, N(2)CH_3_], 4.80 [2H, q, *J* 7.0, N(6)CH_2_CH_3_], 7.97 [1H, d, *J* 8.6, C(7)H], 8.19 [1H, d, *J* 8.6, C(8)H], 8.52 [1H, d, *J* 7.3, C(4)H], 8.67 [1H, d, *J* 7.3, C(3)H], 8.91 [1H, s, C(10)H], 10.13 [1H, s, C(1)H], 10.16 [1H, s, C(9)CHO]; δ_C_ (150.9 MHz, DMSO-*d*_6_): 13.8 [CH_3_, C(5)CH_3_], 15.70 [CH_3_, N(6)CH_2_CH_3_], 15.74 [CH_3_, C(11)CH_3_], 41.1 [CH_2_, N(6)CH_2_CH_3_], 47.0 [CH_3_, N(2)CH_3_], 111.0 (CH, aromatic CH), 112.3 (C, aromatic C), 121.3 (C, aromatic C), 121.4 (CH, aromatic CH), 122.3 (C, aromatic C), 126.6 (C, aromatic C), 127.9 (CH, aromatic CH), 129.8 (CH, aromatic CH), 130.4 (C, aromatic C), 133.2 (CH, aromatic CH), 134.1 (C, aromatic C), 134.7 (C, aromatic C), 144.2 (C, aromatic C), 147.8 C(aromatic C), 148.0 (CH, aromatic CH), 192.6 [C, C(9)CHO]; m/z (ESI^+^): 317 [(M+H)^+^, 100%]; HRMS (ESI^+^): Exact mass calculated for C_21_H_21_N_2_O^+^ 317.1654. Found 317.1662.

### 4.3. 2,6-Diethyl-9-Formyl-5,11-Dimethyl-6H-Pyrido[4,3-b]Carbazol-2-Ium Iodide **21**

Iodoethane (0.044 mL, 0.085 g, 0.546 mmol) was added to a stirred solution of 6-ethyl-5,11-dimethyl-6*H*-pyrido[4,3-*b*]carbazole-9-carbaldehyde **18** (0.150 g, 0.496 mmol) in anhydrous DMF (3.5 mL) and stirred at room temperature for 18 h. The reaction mixture was cooled on ice and cold diethyl ether (3 mL) was added. The resulting precipitate was collected by vacuum filtration and washed with hexane (3 mL) and diethyl ether (5 mL) to give the product as a yellow solid which was dried at 0.2 mbar (0.144 g, 63.5%). m.p. > 300 °C; v_max_/cm^−1^ (KBr): 3046, 3014, 2973, 1671, 1587, 1574, 1242, 839; δ_H_ (600 MHz, DMSO-*d*_6_): 1.48 [3H, t, *J* 7.2, N(6)CH_2_CH_3_], 1.67 [3H, t, *J* 7.2, N(2)CH_2_CH_3_], 3.10 [3H, s, C(5)CH_3_], 3.34 [3H, s, C(11)CH_3_], 4.80 [4H, q, *J* 7.2, N(2)CH_2_CH_3_ and N(6)CH_2_CH_3_], 7.96 [1H, d, *J* 8.6, C(7)H], 8.18 [1H, dd, *J* 8.6, 1.5, C(8)H], 8.69 [2H, d, *J* 7.5, C(3)H, C(4)H], 8.92 [1H, s, C(10)H], 10.15 [1H, s, C(9)CHO], 10.20 [1H, s, C(1)H]; δ_C_ (150.9 MHz, DMSO-*d*_6_): 13.5 [CH_3_, C(5)CH_3_], 15.39 [CH_3_, N(6)CH_2_CH_3_], 15.43 [CH_3_, C(11)CH_3_], 16.7 [CH_3_, N(2)CH_2_CH_3_], 41.2 [CH_2_, N(6)CH_2_CH_3_], 56.0 [CH_2_, N(2)CH_2_CH_3_], 110.8 (CH, aromatic CH), 112.3 (C, aromatic C), 121.4 (C, aromatic C), 121.8 (CH, aromatic CH), 122.2 (C, aromatic C), 126.5 (C, aromatic C), 127.4 (CH, aromatic CH), 130.0 (C, aromatic C), 130.2 (CH, aromatic CH), 131.5 (CH, aromatic CH), 134.2 (C, aromatic C), 134.9 (C, aromatic C), 144.2 (C, aromatic C), 146.3 (CH, aromatic CH), 147.7 (C, aromatic C), 193.2 [C, C(9)CHO]; m/z (ESI^+^): 331 [(M+H)^+^, 100%]; HRMS (ESI^+^): Exact mass calculated for C_22_H_23_N_2_O^+^ 331.1810. Found 331.1804.

### 4.4. 6-Ethyl-9-Formyl-2-Isopropyl-5,11-Dimethyl-6H-Pyrido[4,3-b]Carbazol-2-Ium Iodide **22**

2-Iodopropane (0.055 mL, 0.093 g, 0.546 mmol) was added to a stirred solution of 6-ethyl-5,11-dimethyl-6*H*-pyrido[4,3-*b*]carbazole-9-carbaldehyde **18** (0.150 g, 0.496 mmol) in 1,4-dioxane (8 mL) and stirred at reflux for 16 h. Starting material was still present when the reaction mixture was examined using TLC. Additional 2-iodopropane (0.10 mL, 0.169 g, 0.992 mmol) was added and heated to reflux for a further 24 h. The mixture was cooled on ice and cold diethyl ether (3 mL) was added. The resultant precipitate was filtered and washed with hexane (3 mL) and diethyl ether (5 mL) to give the crude product. Recrystallisation from methanol gave the desired product as a yellow solid, which was dried at 0.2 mbar (0.127 g, 54.4%). m.p. > 300 °C; v_max_/cm^−1^ (KBr): 2977, 2728, 1669, 1586, 1404, 1243, 1098; δ_H_ (300 MHz, DMSO-*d*_6_): 1.48 [3H, t, *J* 7.2, N(6)CH_2_CH_3_], 1.76 [6H, d, *J* 6.8, N(2)CH(CH_3_)_2_], 3.13 [3H, s, C(5)CH_3_], 3.41 [3H, s, C(11)CH_3_], 4.84 [2H, q, *J* 7.2, N(6)CH_2_CH_3_], 5.30 [1H, sept, *J* 6.8, N(2)CH(CH_3_)_2_], 8.00 [1H, d, *J* 8.6, C(7)H], 8.20 [1H, d, *J* 8.6, C(8)H], 8.72 [1H, d, *J* 7.5, C(4)H], 8.80 [1H, d, *J* 7.5, C(3)H], 8.99 [1H, s, C(10)H], 10.17 [1H, s, C(9)CHO], 10.19 [1H, s, C(1)H]; δ_C_ (150.9 MHz, DMSO-*d*_6_): 13.5 [CH_3_, C(5)CH_3_], 15.38 [CH_3_, C(11)CH_3_], 15.41 [CH_3_, N(6)CH_2_CH_3_], 22.7 [2 x CH_3_, N(2)CH(CH_3_)_2_], 41.1 [CH_2_, N(6)CH_2_CH_3_], 63.7 [CH, N(2)CH(CH_3_)_2_] 110.7 (CH, aromatic CH), 112.3 (C, aromatic C), 121.3 (C, aromatic C), 122.1 (CH, aromatic CH), 122.2 (C, aromatic C), 126.6 (C, aromatic C), 127.4 (CH, aromatic CH), 129.0 (CH, aromatic CH), 129.9 (C, aromatic C), 130.2 (CH, aromatic CH), 134.3 (C, aromatic C), 135.3 (C, aromatic C), 144.2 (C, aromatic C), 145.2 (CH, aromatic CH), 147.8 (C, aromatic C), 193.2 [C, C(9)CHO]; m/z (ESI^+^): 345 [(M+H)^+^, 100%]; HRMS (ESI^+^): Exact mass calculated for C_23_H_25_N_2_O^+^ 345.1967. Found 345.1958.

### 4.5. 2-(5-Cyanopentyl)-6-Ethyl-9-Formyl-5,11-Dimethyl-6H-Pyrido[4,3-b]Carbazol-2-Ium Bromide **23**


6-Bromohexanenitrile (0.052 mL, 69.7 mg, 0.396 mmol) was added to a stirred suspension of 6-ethyl-5,11-dimethyl-6*H*-pyrido[4,3-*b*]carbazole-9-carbaldehyde **18** (100 mg, 0.331 mmol) in dimethylformamide (5 mL) and heated to 120 °C for four hours. The reaction mixture was cooled on ice and cold diethyl ether (5 mL) added. The resultant precipitate was collected by filtration and washed with hexane to give the product as a yellow solid which was dried at 0.2 mbar (123 mg, 77.9%). m.p. 246–248 °C; ν_max_/cm^−1^ (KBr): 3374, 2932, 2240, 1680, 1642, 1580, 1460, 1395, 1357, 1239, 814; δ_H_ (500 MHz, DMSO-*d*_6_): 1.40 – 1.52 [5H, m, N(6)CH_2_CH_3_, C(3′)H_2_], 1.62 – 1.71 [2H, m, C(4′)H_2_], 2.03 – 2.11 [2H, m, C(2′)H_2_], 2.55 [2H, t, *J* 7.0, C(5′)H_2_], 3.09 [3H, s, C(5)CH_3_], 3.34 [3H, s, C(11)CH_3_], 4.75 – 4.83 [4H, m, N(6)CH_2_CH_3_, C(1′)H_2_], 7.96 [1H, d, *J* 8.6, C(7)H], 8.17 [1H, d, *J* 8.4, C(8)H], 8.66 [1H, d, *J* 7.2, C(4)H], 8.71 [1H, d, *J* 7.2, C(3)H], 8.91 [1H, s, C(10)H], 10.14 [1H, s, C(9)CHO], 10.23 [1H, s, C(1)H]; δ_C_ (125.8 MHz, DMSO-*d*_6_): 13.8 [CH_3_, C(5)CH_3_], 15.7 [CH_3_, N(6)CH_2_CH_3_], 15.9 [CH_3_, C(11)CH_3_], 16.5 [CH_2_, C(5′)H_2_], 24.7 [CH_2_, C(4′)H_2_], 25.2 [CH_2_, C(3′)H_2_], 30.5 [CH_2_, C(2′)H_2_], 41.1 [CH_2_, N(6)CH_2_CH_3_], 59.7 [CH_2_, C(1′)H_2_], 111.1 (CH, aromatic CH), 112.4 (C, aromatic C), 121.1 (C, CN), 121.5 (C, aromatic C), 121.8 (CH, aromatic CH), 122.3 (C, aromatic C), 126.8 (C, aromatic C), 127.9 (CH, aromatic CH), 129.8 (CH, aromatic CH), 130.5 (C, aromatic C), 132.2 (CH, aromatic CH), 134.5 (C, aromatic C), 134.9 (C, aromatic C), 144.3 (C, aromatic C), 147.3 (CH, aromatic CH), 147.8 (C, aromatic C), 192.6 [C, C(9)CHO]; m/z (ESI^+^): 398 [(M)^+^, 100%]; HRMS (ESI^+^): Exact mass calculated for C_26_H_28_N_3_O^+^ 398.2232. Found 398.2237.

### 4.6. 6-Ethyl-9-Formyl-5,11-Dimethyl-2-Tetradecyl-6H-Pyrido[4,3-b]Carbazol-2-Ium Bromide **24**

1-Bromotetradecane (0.13 mL, 0.120g, 0.434 mmol) was added to a stirred solution of 6-ethyl-5,11-dimethyl-6*H*-pyrido[4,3-*b*]carbazole-9-carbaldehyde **18** (0.119g, 0.394 mmol) in anhydrous DMF (8 mL) and stirred at reflux for 16 h. The mixture was cooled on ice and cold diethyl ether was added (3 mL). The resultant precipitate was filtered and washed with hexane (3 mL) and diethyl ether (5 mL) to give the product as a yellow solid which was dried at 0.2 mbar (0.156 g, 68.5%). m.p. 194–197 °C; v_max_/cm^−1^ (KBr): 2922, 2852, 1674, 1586, 1356, 1243, 1101, 810; δ_H_ (300 MHz, DMSO-*d*_6_): 0.83 [3H, t, *J* 6.6, C(14′)H], 1.12-1.41 [22H, m, (3′)H_2_-C(13′)H_2_], 1.48 [3H, t, 7.1, N(6)CH_2_CH_3_], 1.96-2.10 [2H, m, C(2′)H], 3.11 [3H, s, C(5)CH_3_], 3.37 [1H, s, C(11)CH_3_], 4.76 [2H, t, *J* 7.5, C(1′)H], 4.83 [2H, q, *J* 7.1, N(6)CH_2_CH_3_], 7.99 [1H, d, *J* 8.6, C(7)H], 8.20 [1H, dd, *J* 8.6, 1.4, C(8)H], 8.66 [1H, d, *J* 7.6, C(4)H], 8.72 [1H, d, *J* 7.6, C(3)H], 8.96 [1H, d, *J* 1.2, C(10)H], 10.17 [1H, s, C(9)CHO], 10.24 [1H, s, C(1)H]; δ_C_ (150.9 MHz, DMSO-*d*_6_): 13.6 [CH_3_, C(5)CH_3_], 14.4 [CH_3_, C(14′)H_3_], 15.7 [CH_3_, N(6)CH_2_CH_3_], 15.9 [CH_3_, C(11)CH_3_], 22.5 (CH_2_), 26.1 (CH_2_), 29.0 (CH_2_), 29.1 (CH_2_), 29.3 (CH_2_), 29.4 (CH_2_), 29.5 (2 x CH_2_), 29.5 (2 x CH_2_), 31.4 (CH_2_), 31.7 [CH_2_, C(2′)H_2_], 41.0 [CH_2_, N(6)CH_2_CH_3_], 59.9 [CH_2_, C(1′)H_2_], 101.7 (CH, aromatic CH), 112.2 (C, aromatic C), 121.3 (C, aromatic C), 121.7 (CH, aromatic CH), 122.1 (C, aromatic C), 126.5 (C, aromatic C), 127.7 (CH, aromatic CH), 129.6 (CH, aromatic CH), 130.3 (C, aromatic C), 132.1 (CH, aromatic CH), 134.2 (C, aromatic C), 134.7 (C, aromatic C), 144.0 (C, aromatic C), 147.0 (CH, aromatic CH), 147.5 (C, aromatic C), 192.4 [C, C(9)CHO]; m/z (ESI^+^): 499 [(M+H)^+^, 100%]; HRMS (ESI^+^): Exact mass calculated for C_34_H_47_N_2_O^+^ 499.3688. Found 499.3691.

### 4.7. 6-Ethyl-9-Formyl-5,11-Dimethyl-2-Octadecyl-6H-Pyrido[4,3-b]Carbazol-2-Ium Bromide **25**


1-Bromooctadecane (140 mg, 0.42 mmol) was added to a stirred suspension of 6-ethyl-5,11-dimethyl-6*H*-pyrido[4,3-*b*]carbazole-9-carbaldehyde **18** (105 mg, 0.35 mmol) in dimethylformamide (5 mL) and heated to 120 °C for four hours. The reaction mixture was cooled on ice and cold diethyl ether (5 mL) added. The resultant precipitate was collected by filtration and washed with hexane to give the product as a yellow solid which was dried at 0.2 mbar (182.5 mg, 82.0%). m.p. 199–203 °C; ν_max_/cm^−1^ (KBr): 3418, 2921, 2851, 1680, 1581, 1459, 1399, 1357, 1242, 1101, 810; δ_H_ (500 MHz, DMSO-*d*_6_): 0.83 [3H, t, *J* 6.6, C(18′)H_3_], 1.10–1.40 [30H, m, C(3′)H_2_–C(17′)H_2_], 1.48 [3H, t, *J* 6.8, N(6)CH_2_CH_3_], 1.99–2.08 [2H, m, C(2′)H_2_], 3.11 [3H, s, C(5)CH_3_], 3.37 [3H, s, C(11)CH_3_], 4.76 [2H, t, *J* 6.8, C(1′)H_2_], 4.82 [2H, q, *J* 6.7, N(6)CH_2_CH_3_], 7.99 [1H, d, *J* 8.5, C(7)H], 8.20 [1H, d, *J* 8.4, C(8)H], 8.66 [1H, d, *J* 7.1, C(4)H], 8.72 [1H, d, *J* 7.0, C(3)H], 8.96 [1H, s, C(10)H], 10.16 [1H, s, C(9)CHO], 10.23 [1H, s, C(1)H]; δ_C_ (125.8 MHz, DMSO-*d*_6_): 13.8 [CH_3_, C(5)CH_3_], 14.4 [CH_3_, C(18′)H_3_], 15.7 [CH_3_, N(6)CH_2_CH_3_], 15.9 [CH_3_, C(11)CH_3_], 22.5 (CH_2_), 26.1 (CH_2_), 28.9 (CH_2_), 29.1 (CH_2_), 29.26 (CH_2_), 29.32 (CH_2_), 29.5 (8 × CH_2_), 31.3 [CH_2_, C(2′)H_2_], 31.7 (CH_2_), 41.1 [CH_2_, N(6)CH_2_CH_3_], 60.1 [CH_2_, C(1′)H_2_], 111.1 (CH, aromatic CH), 112.5 (C, aromatic C), 121.5 (C, aromatic C), 121.8 (CH, aromatic CH), 122.4 (C, aromatic C), 126.8 (C, aromatic C), 127.9 (CH, aromatic CH), 129.9 (CH, aromatic CH), 130.6 (C, aromatic C), 132.2 (CH, aromatic CH), 134.5 (C, aromatic C), 135.0 (C, aromatic C), 144.4 (C, aromatic C), 147.3 (CH, aromatic CH), 147.8 (C, aromatic C), 192.6 [C, C(9)CHO]; m/z (ESI^+^): 555 [(M)^+^, 100%]; HRMS (ESI^+^): Exact mass calculated for C_38_H_55_N_2_O^+^ 555.4314. Found 555.4293.

### 4.8. 2-Ethyl-9-Formyl-6-Isopropyl-5,11-Dimethyl-6H-Pyrido[4,3-b]Carbazol-2-Ium Iodide **26**


Iodoethane (0.05 mL, 97 mg, 0.62 mmol) was added to a suspension of 6-isopropyl-5,11-dimethyl-6*H*-pyrido[4,3-*b*]carbazole-9-carbaldehyde **19** (99 mg, 0.31 mmol) in dimethylformamide (5 mL) and stirred at room temperature for 16 h. Thin layer chromatography analysis indicated a large amount of starting material was still present. Additional iodoethane (0.05 mL, 97 mg, 0.62 mmol) was added and the reaction was stirred at room temperature for a further 24 h. The mixture was cooled on ice and diethyl ether (5 mL) added. The resultant precipitate was filtered, washed with hexane and dried by heating under reduced pressure (110 °C, 0.2 mbar) to give the product as a yellow solid (116.1 mg, 78.8%). m.p. > 300 °C without melting; ν_max_/cm^−1^ (KBr): 2974, 1675, 1579, 1451, 1397, 1310, 1246, 1209, 1100, 807; δ_H_ (300 MHz, DMSO-*d*_6_): 1.65 [3H, t, *J* 7.2, N(2)CH_2_CH_3_], 1.73 [6H, d, *J* 7.0, N(6)CH(CH_3_)_2_], 3.08 [3H, s, C(5)CH_3_], 3.40 [3H, s, C(11)CH_3_], 4.81 [2H, q, *J* 7.2, N(2)CH_2_CH_3_], 5.61 [1H, sept, *J* 6.8, N(6)CH(CH_3_)_2_], 8.14–8.19 [2H, m, C(7)H, C(8)H], 8.65 – 8.72 [2H, m, C(3)H, C(4)H], 9.02 [1H, s, C(10)H], 10.20 [1H, s, C(9)CHO], 10.24 [1H, s, C(1)H]; δ_C_ (150.9 MHz, DMSO-*d*_6_): 15.7 [CH_3_, C(5)CH_3_], 15.8 [CH_3_, C(11)CH_3_], 17.1 [CH_3_, N(2)CH_2_CH_3_], 21.3 [2 × CH_3_, N(6)CH(CH_3_)_2_], 50.4 [CH, N(6)CH(CH_3_)_2_], 55.8 [CH_2_, N(2)CH_2_CH_3_], 112.8 (C, aromatic C), 114.6 (CH, aromatic CH), 121.7 (C, aromatic C), 122.2 (CH, aromatic CH), 124.0 (C, aromatic C), 126.9 (C, aromatic C), 128.1 (CH, aromatic CH), 129.0 (CH, aromatic CH), 130.2 (C, aromatic C), 132.0 (CH, aromatic CH), 134.0 (C, aromatic C), 135.5 (C, aromatic C), 146.4 (C, aromatic C), 146.8 (C, aromatic C), 147.2 (CH, aromatic CH), 192.8 [C, C(9)CHO]; m/z (ESI^+^): 345 [(M)^+^, 100%]; HRMS (ESI^+^): Exact mass calculated for C_23_H_25_N_2_O^+^ 345.1967. Found 345.1965.

### 4.9. 2-(5-Cyanopentyl)-9-Formyl-6-Isopropyl-5,11-Dimethyl-6H-Pyrido[4,3-b]Carbazol-2-Ium Bromide **27**

6-Bromohexanenitrile (0.04 mL, 0.049g, 0.278 mmol) was added to a stirred suspension of 6-isopropyl-5,11-dimethyl-6*H*-pyrido[4,3-*b*]carbazole-9-carbaldehyde **19** (0.080 g, 0.253 mmol) in anhydrous DMF (3 mL) and stirred at 120 °C for 16 h. TLC showed the presence of starting material so a further portion of 6-bromohexanenitrile (0.018 mL, 0.024 g, 0.139 mmol) was added and the reaction was heated to 120 °C for a further 24 h. The reaction mixture was cooled on ice and cold diethyl ether was added (3 mL). The resultant precipitate was filtered and washed with hexane (3 mL) and diethyl ether (5 mL) to give the product as a yellow solid which was dried at 0.2 mbar (0.089 g, 71.9%). m.p. 253–255 °C; v_max_/cm^−1^ (KBr): 3383, 2937, 2241, 1677, 1585, 1399, 1251, 1101; δ_H_ (600 MHz, DMSO-*d*_6_): 1.42-1.51 [2H, m, C(3′)H_2_], 1.67 [2H, quin, *J* 7.4, C(4′)H_2_], 1.73 [6H, d, *J* 6.9, N(6)CH(CH_3_)_2_], 2.08 [2H, quin, *J* 7.4, C(2′)H_2_], 2.55 [2H, t, *J* 7.4, C(5′)H_2_], 3.08 [3H, s, C(5)CH_3_], 3.40 [3H, s, C(11)CH_3_], 4.80 [2H, t, *J* 7.4, C(1′)H_2_], 5.61 [1H, sept, *J* 6.9, N(6)CH(CH_3_)_2_], 8.17 [2H, s, C(7)H, C(8)H], 8.71 [2H, s, C(3)H, C(4)H], 9.02 [1H, s, C(10)H], 10.20 [1H, s, C(9)CHO], 10.28 [1H, s, C(1)H]; δ_C_ (150.9 MHz, DMSO-*d*_6_): 15.7 [CH_3_, C(5)CH_3_], 15.9 [CH_3_, C(11)CH_3_], 16.5 [CH_2_, C(5′)H_2_], 21.3 [2 x CH_3_, N(6)CH(CH_3_)_2_], 24.7 [CH_2_, C(4′)H_2_], 25.1 [CH_2_, C(3′)H_2_], 30.5 [CH_2_, C(2′)H_2_], 50.4 [CH, N(6)CH(CH_3_)_2_], 59.7 [CH_2_, C(1′)H_2_], 112.8 (C, aromatic C), 114.6 (CH, aromatic CH, one of C(7)H or C(8)H), 121.1 (C, CN), 121.8 (C, aromatic C), 122.0 [CH, C(1)H], 124.1 (C, aromatic C), 127.0 (C, aromatic C), 128.1 (CH, aromatic CH), 129.1 (CH, aromatic CH, one of C(7)H or C(8)H), 130.3 (C, aromatic C), 132.3 [CH, C(3)H], 134.1 (C, aromatic C), 135.6 (C, aromatic C), 146.5 (C, aromatic C), 146.8 (C, aromatic C), 147.4 (CH, aromatic CH), 192.7 [C(9)CHO]; m/z (ESI^+^): 412 [(M+H)^+^, 100%]; HRMS (ESI^+^): Exact mass calculated for C_27_H_30_N_3_O^+^ 412.2389. Found 412.2393.

### 4.10. 9-Formyl-6-Isopropyl-5,11-Dimethyl-2-Tetradecyl-6H-Pyrido[4,3-b]Carbazol-2-Ium Bromide **28**


1-Bromotetradecane (0.11 mL, 102 mg, 0.368 mmol) was added to a stirred suspension of of 6-isopropyl-5,11-dimethyl-6*H*-pyrido[4,3-*b*]carbazole-9-carbaldehyde **19** (100 mg, 0.316 mmol) and heated to 120 °C for 4 h. Following thin layer chromatography analysis, further 1-bromotetradecane (0.06 mL, 56 mg, 0.201 mmol) was added and the reaction was heated to 120 °C for 16 h. The reaction was cooled on ice and cold diethyl ether (5 mL) added. The resultant precipitate was filtered, washed with hexane and dried at 0.2 mbar to give the product as a yellow solid (149 mg, 79.7%). m.p. 225–227 °C; ν_max_/cm^−1^ (KBr): 3421, 2924, 2852, 1682, 1583, 1461, 1396, 1245, 1102; δ_H_ (500 MHz, DMSO-*d*_6_): 0.83 [3H, t, *J* 6.9, C(14′)H_3_], 1.12–1.39 [22H, m, C(3′)H_2_–C(13′)H_2_], 1.73 [6H, d, *J* 6.9, N(6)CH(CH_3_)_2_], 1.99–2.09 [2H, m, C(2′)H_2_], 3.07 [3H, s, C(5)CH_3_], 3.39 [3H, s, C(11)CH_3_], 4.78 [2H, t, *J* 7.3, C(1′)H_2_], 5.60 [1H, sept, *J* 6.7, N(6)CH(CH_3_)_2_], 8.16 [2H, br s, C(7)H, C(8)H], 8.69 [2H, br s, C(3)H, C(4)H], 9.00 [1H, s, C(10)H], 10.19 [1H, s, C(9)CHO], 10.26 [1H, s, C(1)H]; δ_C_ (125.8 MHz, DMSO-*d*_6_): 14.4 [CH_3_, C(14′)H_3_], 15.7 [CH_3_, C(5)CH_3_], 15.9 [CH_3_, C(11)CH_3_], 21.3 [2 × CH_3_, N(6)CH(CH_3_)_2_], 22.5 (CH_2_), 26.0 (CH_2_), 28.9 (CH_2_), 29.1 (CH_2_), 29.27 (CH_2_), 29.33 (CH_2_), 29.42 (CH_2_), 29.44 (CH_2_), 29.5 (2 × CH_2_), 31.3 [CH_2_, C(2′)H_2_], 31.7 (CH_2_), 50.4 [CH, N(6)CH(CH_3_)_2_], 60.2 [CH_2_, C(1′)H_2_], 112.8 (C, aromatic C), 114.6 (CH, aromatic CH), 121.8 (C, aromatic C), 122.1 (CH, aromatic CH), 124.1 (C, aromatic C), 127.0 (C, aromatic C), 128.0 (CH, aromatic CH), 129.1 (CH, aromatic CH), 130.3 (C, aromatic C), 132.3 (CH, aromatic CH), 134.0 (C, aromatic C), 135.5 (C, aromatic C), 146.4 (C, aromatic C), 146.8 (C, aromatic C), 147.3 (CH, aromatic CH), 192.7 [C, C(9)CHO]; m/z (ESI^+^): 513 [(M)^+^, 100%]; HRMS (ESI^+^): Exact mass calculated for C_35_H_49_N_2_O^+^ 513.3845. Found 513.3844.

### 4.11. 9-Formyl-6-Isopropyl-5,11-Dimethyl-2-Octadecyl-6H-Pyrido[4,3-b]Carbazol-2-Ium Bromide **29**

1-Bromooctadecane (0.10 mL, 0.094 g, 0.278 mmol) was added to a stirred suspension of 6-isopropyl-5,11-dimethyl-6*H*-pyrido[4,3-*b*]carbazole-9-carbaldehyde **19** (0.080 g, 0.253 mmol) in anhydrous DMF (3 mL) and stirred at 120 °C for 16 h. A red-brown solution formed on heating. The reaction mixture was cooled on ice and cold diethyl ether was added (3 mL). The resultant precipitate was filtered and washed with hexane (3 mL) and diethyl ether (5 mL) to give the product as a yellow solid which was dried at 0.2 mbar (0.128 g, 78.1%). m.p. 138-141 °C; v_max_/cm^−1^ (KBr): 2953, 2919, 2850, 1675, 1592, 1251, 1101; δ_H_ (600 MHz, DMSO-*d*_6_): 0.83 [3H, t, *J* 7.1, C(18′)H_3_], 1.11–1.39 [30H, m, C(3′)H_2_-C(17′)H_2_], 1.73 [6H, d, *J* 7.0, N(6)CH(CH_3_)_2_], 2.03 [2H, quin, *J* 7.1, C(2′)H_2_], 3.08 [3H, s, C(5)CH_3_], 3.39 [3H, s, C(11)CH_3_], 4.78 [2H, t, *J* 7.5, C(1′)H_2_], 5.60 [1H, sept, *J* 7.0, N(6)CH(CH_3_)_2_], 8.16 [2H, s, C(7)H, C(8)H], 8.69 [2H, s, C(3)H, C(4)H], 8.99 [1H, s, C(10)H], 10.18 [1H, s, C(9)CHO], 10.26 [1H, s, C(1)H]; δ_C_ (150.9 MHz, DMSO-*d*_6_): 14.4 [CH_3_, C(18′)H_3_], 15.7 [CH_3_, C(5)CH_3_], 15.9 [CH_3_, C(11)CH_3_], 21.3 [2 x CH_3_, N(6)CH(CH_3_)_2_], 22.5 (CH_2_), 26.0 (CH_2_), 28.9 (CH_2_), 29.1 (CH_2_), 29.24 (CH_2_), 29.30 (CH_2_), 29.4 (CH_2_), 29.5 (6 x CH_2_), 31.3 [CH_2_, C(2′)H_2_], 31.7 (CH_2_), 40.4 (CH_2_), 50.4 [CH, N(6)CH(CH_3_)_2_], 60.2 [CH_2_, C(1′)H_2_], 112.8 (C, aromatic C), 114.7 (CH, aromatic CH), 121.9 (C, aromatic C), 122.1 (CH, aromatic CH), 124.1 (C, aromatic C), 127.1 (C, aromatic C), 128.1 (CH, aromatic CH), 129.2 (CH, aromatic CH), 130.3 (C, aromatic C), 132.3 (CH, aromatic CH), 134.1 (C, aromatic C), 135.6 (C, aromatic C), 146.5 (C, aromatic C), 146.9 (C, aromatic C), 147.3 (CH, aromatic CH), 192.7 [C, C(9)CHO]; m/z (ESI^+^): 596 [(M+H)^+^, 100%]; HRMS (ESI^+^): Exact mass calculated for C_39_H_57_N_2_O^+^ 569.4471. Found 569.4462.

### 4.12. 5,10,11-Trimethyl-10H-Pyrido[3,4-b]Carbazole **31**


Sodium hydride (134 mg, 60% dispersion in mineral oil, equivalent to 80 mg sodium hydride, 3.3 mmol) was suspended in dimethylformamide (5 mL) under nitrogen and 5,11-dimethyl-10*H*-pyrido[3,4-*b*]carbazole **30** (274 mg, 1.11 mmol) added portionwise, forming a deep red suspension which was stirred at room temperature for 45 min. Iodomethane (0.07 mL, 160 mg, 1.12 mmol) in dimethylformamide (3 mL) was added dropwise and the reaction was stirred overnight. The reaction mixture was cooled on ice and water (30 mL) added. The resultant yellow precipitate was filtered and washed with excess water to give the product which was used without further purification (276 mg, 95.5%). m.p. 165–167 °C (Lit. m.p. 157–159) [45]; ν_max_/cm^−1^ (KBr): 3047, 3016, 2926, 1591, 1481, 1388, 1356, 1329, 1304, 1234, 989, 741; δ_H_ (300 MHz, DMSO-*d*_6_): 3.02 [3H, s, C(11)CH_3_], 3.13 [3H, s, C(5)CH_3_], 4.09 [3H, s, N(10)CH_3_], 7.23–7.33 [1H, m, C(7)H], 7.56–7.64 [2H, m, C(8)H, C(9)H], 8.07 [1H, d, *J* 6.0, C(4)H], 8.34 [1H, d, *J* 7.9, C(6)H], 8.42 [1H, d, *J* 6.0, C(3)H], 9.63 [1H, s, C(1)H]; m/z (ESI^+^): 261 [(M+H)^+^, 100%]. 

### 4.13. 5,10,11-Trimethyl-10H-Pyrido[3,4-b]Carbazole-7-Carbaldehyde **32**

5,10,11-Trimethyl-10*H*-pyrido[3,4-*b*]carbazole **31** (920 mg, 3.53 mmol) was stirred in trifluoroacetic acid (50 mL) and hexamethylenetetramine (4.95 g, 35.3 mmol) added portionwise. The solution was refluxed for 30 min, cooled to room temperature and concentrated to one third volume under reduced pressure. Water (250 mL) was added and pH was adjusted to 6 with sodium bicarbonate on ice (Caution: vigorous reaction). Dichloromethane – methanol 90:10 (1 × 200mL) was added and an emulsion formed which resolved upon stirring overnight. The organic layer was separated and the aqueous layer extracted with further dichloromethane – methanol 90:10 (2 × 100 mL). Combined organic extracts were washed with water (2 × 100 mL) and brine (1 × 100 mL), dried over magnesium sulfate and solvent removed under reduced pressure to give the product as a yellow solid (916 mg, 89.8%). m.p. 169–171 °C; v_max_/cm^−1^ (KBr):3417, 2924, 2855, 1678, 1585, 1461, 1360, 1303, 1203, 1097, 802; δ_H_ (500 MHz, DMSO-*d*_6_): 2.79 [3H, s, C(11)CH_3_], 2.94 [3H, s, C(5)CH_3_], 3.93 [3H, s, N(10)CH_3_], 7.57 [1H, d, *J* 8.5, C(9)H], 7.94 [1H, d, *J* 5.9, C(4)H], 7.98 [1H, d, *J* 8.5, C(8)H], 8.40 [1H, d, *J* 5.7, C(3)H], 8.50 [1H, s, C(6)H], 9.53 [1H, s, C(1)H], 9.98 [1H, s, C(7)CHO]; δ_C_ (125.8 MHz, DMSO-*d*_6_): 13.3 [CH_3_, C(5)CH_3_], 14.8 [CH_3_, C(11)CH_3_], 34.3 [CH_3_, N(10)CH_3_], 109.5 (CH, aromatic CH), 113.0 (C, aromatic C), 116.8 (CH, aromatic CH), 122.3 (C, aromatic C), 125.7 (C, aromatic C), 126.3 (C, aromatic C), 126.7 (C, aromatic C), 127.3 (CH, aromatic CH), 128.7 (C, aromatic C), 128.9 (CH, aromatic CH), 129.4 (C, aromatic C), 139.1 (C, aromatic C), 139.8 (CH, aromatic CH), 148.7 (C, aromatic C), 149.3 (CH, aromatic CH), 194.1 [C, C(7)CHO]; m/z (ESI^+^): 289 [(M + H)^+^,100%]; HRMS (ESI^+^): Exact mass calculated for C_19_H_17_N_2_O^+^ 289.1341. Found 289.1341.

### 4.14. 7-Formyl-2,5,11-Trimethyl-10H-Pyrido[3,4-b]Carbazol-2-Ium Iodide **33**


5,11-Dimethyl-10*H*-pyrido[3,4-*b*]carbazole **30** (1.72 g, 6.98 mmol) was dissolved in trifluoroacetic acid (100 mL) and hexamethylenetetramine (9.79 g, 69.8 mmol) gradually added over five minutes. The mixture was heated to reflux for 40 min and upon cooling, concentrated to half volume under reduced pressure. The mixture was placed on ice and water (200 mL) added. The solution was neutralised with solid sodium bicarbonate (Caution: vigorous reaction) and the resultant orange precipitate isolated by filtration. Column chromatography on neutral alumina eluting with dichloromethane–methanol 95:5 gave the product 5,11-Dimethyl-10*H*-pyrido[3,4-*b*]carbazole-7-carbaldehyde (1.38 g, 72.3%). m.p. 294–296 °C (Lit. m.p. 270–271 °C)[45]; ν_max_/cm^−1^ (KBr): 3029, 2864, 1676, 1600, 1471, 1400, 1380, 1317, 1290, 1217, 1202, 1113, 1015, 807; δ_H_ (300 MHz, DMSO-*d*_6_): 2.94 [3H, s, C(11)CH_3_], 3.16 [3H, s, C(5)CH_3_], 7.66 [1H, d, *J* 8.4, C(9)H], 8.06 [1H, dd, *J* 8.4, 1.3, C(8)H], 8.14 [1H, d, *J* 6.0, C(4)H], 8.46 [1H, d, *J* 6.1, C(3)H], 8.86 [1H, s, C(6)H], 9.59 [1H, br s, C(1)H], 10.09 [1H, s, C(7)CHO], 11.90 [1H, s, N(10)H]; m/z (ESI^+^): 275 [(M + H)^+^, 100%].

5,11-Dimethyl-10*H*-pyrido[3,4-*b*]carbazole-7-carbaldehyde (188 mg, 0.68 mmol) was then suspended in dimethylformamide (5 mL) and iodomethane (0.06 mL, 136 mg, 0.96 mmol) added. The mixture was stirred for 16 h at room temperature, cooled on ice and cold diethyl ether (5 mL) added. The resultant precipitate was filtered under nitrogen, washed with hexane and dried under vacuum (0.2 mbar) to give the product as a red solid (254 mg, 89.7%). m.p. > 300 °C without melting (Lit. m.p. > 300 °C without melting) [45]; ν_max_/cm^−1^ (KBr): 3418, 3142, 3008, 2838, 2749, 1677, 1638, 1621, 1600, 1467, 1408, 1385, 1318, 1303, 1241, 1199, 1113; δ_H_ (300 MHz, DMSO-*d*_6_): 2.98 [3H, s, C(11)CH_3_], 3.18 [3H, s, C(5)CH_3_], 4.54 [3H, s, N(2)CH_3_], 7.69 [1H, d, *J* 8.5, C(9)H], 8.11 [1H, dd, *J* 8.5, 0.9, C(8)H], 8.49 [1H, d, *J* 7.0, C(4)H], 8.75 [1H, d, *J* 7.0, C(3)H], 8.86 [1H, s, C(6)H], 9.99 [1H, s, C(1)H], 10.11 [1H, s, C(7)CHO], 12.34 [1H, s, N(10)H]; m/z (ESI^+^): 289 [(M)^+^, 100%].

### 4.15. Phytophthora infestans Mycelial Growth Assays

*Phytophthora infestans* stain 88069 (mating type A1) was originally obtained from Professor Sophien Kamoun (The Sainsbury Laboratory, Norwich, UK) and was cultured and maintained on RyeA agar at 20 °C in the dark [46]. For mycelial growth assay, compounds were dissolved in DMSO in a vial and 15mL Rye B agar added, at a temperature of approximately 50 °C (the final concentration of DMSO was 0.1%). The vial was shaken to distribute the contents and the mixture was then poured into a clean, sterile 10 cm petri dish. A 10 mm plug of *P. infestans* mycelium was placed in the centre of the dish, which was then sealed with parafilm. The experiments were carried out in triplicate at 25 μM (unless otherwise stated), stored at 20 °C in the dark and checked periodically for mycelial growth. 

### 4.16. Phytophthora infestans Zoosporogenesis and Zoospore Motility Assays 

Zoosporogenesis was assayed as described by Lu et al. [38]. Following two weeks of culture on Rye B agar, sporangia were collected from plates in 5 mL of modified Petri’s solution (MPS, 5 mM CaCl_2_, 1 mM MgSO_4_, 1 mM KH_2_PO_4_ and 0.8 mM KCl), by scraping with a scalpel blade and were transferred to a 50 mL sterile tube. Plates were washed with another 5 mL of MPS and this was combined with the first 5 mL. To initiate zoospore release, sporangia were placed on ice, in the dark, for 3h. Sporangia or zoospores were counted by brightfield microscopy using a 10× objective and a haemocytometer, in duplicate. 

The motility of harvested zoospores was analysed essentially as described by Appiah et al. [47]. In brief, test compounds were added to 100 μL of zoospores, which were immediately transferred to a 100 mm^2^ square chamber on a microscope chamber, formed by extruding petroleum jelly from a 5 mL syringe. This chamber was sealed with a 0-thickness glass cover-slip, was inverted and placed on a 100× magnification, 1.3 numerical aperture oil-immersion objective of an Olympus IX51 microscope. Using brightfield illumination, the motion of the zoospores within this chamber were recorded for 10 s, at a rate of 10 frames/s, using a Hamamatsu ORCA ER CCD video camera ((Hamamatsu Photonics Ltd., Hertfordshire, UK) and Andor IQ 1.9 acquisition software (Belfast, Northern Ireland). Video recordings were exported as multi-dimensional TIFF files and zoospore trajectories were plotted using the Manual Tracking plugin of ImageJ (https://imagej.nih.gov/ij/plugins/track/Manual%20Tracking%20plugin.pdf). Cell traces were exported in Microsoft Excel format and were analysed using the Chemotaxis and Migration Tool from Ibidi (https://ibidi.com/img/cms/resources/AG/FL_AG_035_Chemotaxis_150dpi.pdf). For each zoospore, this generated numerical data on the velocity, accumulated distance travelled, Euclidean (straight-line from origin to furthest point) distance travelled and directionality (“straightness” of travel, where 1 represents completely linear travel and 0, completely circular). 

### 4.17. XTT Reduction Assay

Human embryonic kidney-293T (HEK-293T) cells were obtained from LGC Standards (Middlesex, UK) and were maintained in Dulbecco’s Modified Eagles Medium containing 10% foetal bovine serum, 100 units/mL penicillin and 100 μg/mL streptomycin, at 37 °C in a humidified atmosphere of 5% CO_2_/95% air. For assays, cells were subcultured on 96-well microtitre plates at a density of 5 × 10^4^ cells/well, in a volume of 50 mL. Following overnight culture, 50 mL of each test compound at twice their final concentration (50 mM for all apart from cinnamaldehyde (2 mM) and Mancozeb (200 M)) was added and incubated for another 24 h. Cells were incubated with 100 mL XTT reagents (Sigma-Aldrich, Ireland) for an additional 4 h, after which, the absorbance of the extracellular formazan reduction product was measured at a wavelength of of 490 nm, with a reference wavelength of 675 nm [41,42], using a microtitre plate spectrophotometer.

### 4.18. Statistical Methods

Numerical data were statistically compared using one-way analysis of variance (ANOVA) and Tukey’s post-hoc test (using ezANOVA software, available at: https://people.cas.sc.edu/rorden/ezanova/index.html.) A probability threshold of *p* < 0.05 was taken as statistically significant. Generation of histograms and linear regression analyses were performed using Microsoft Excel. Concentration-growth inhibition data were analysed using GraphPad Prism software version 4.03 (San Diago, CA, USA).

## 5. Conclusions

It is evident from the data that both mammalian cell viability and *P. infestans* inhibitory effects are impacted by discreet modifications of the ellipticine scaffold. The aldehyde functionality on ellipticine has been shown to improve the inhibitory effects on *P. infestans* growth, and remarkably the aldehyde does not consistently result in mammalian toxicity which is a significant finding for future studies. 

The XTT cytotoxicity assay has highlighted the importance of substitution as without this, both ellipticine and isoellipticine parent compounds (**15**) and (**30**) were amongst the most highly toxic of all the compounds tested. This is compounded by the toxicity of compounds **24** and **28** with a consistent C-14 alkyl chain at the 2-position and it is of interest that if the chain is extended or reduced the toxicity is muted and will be probed further in future studies. 

The mycelium growth assay provided information at day five, nine and thirteen days of *P. infestans* growth. It is evident that compounds **20**–**22** demonstrate the greatest inhibitory effects on *P. infestans* with minimal mycelium growth after thirteen days of incubation and no zoosporogenisis. Compound **21** emerges as the most promising candidate, showing low toxicity at 25 μM (75% of control cell growth), no formation of zoospores, more potent inhibition than cinnamaldehyde (**7**) and greater inhibition than that of Mancozeb (**1**), a currently marketed pesticide. In addition, given the extensive effect of the compound it is likely that lower concentrations could be used to the same *P. infestans* growth inhibitory effect but lower toxicity and this will the scope of our future work in addition to small scale field work. This study has resulted in a promising new series of ellipticinium salts as candidates for *P. infestans* treatment and control that warrant further study into their potential as a marketable crop protectant.

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
