# Peer review of "Synthesis and Evaluation of Novel Ellipticines and Derivatives as Inhibitors of *Phytophthora infestans"

_pathogens, 2020, doi:10.3390/pathogens9070558_

Round 1
Reviewer 1 Report
Pathogens-843287
Synthesis and evaluation of novel ellipticines and derivatives as inhibitors of Phytophthora infestans
The manuscript describes a laboratory test experiment on synthesis and evaluation of novel ellipticines and derivatives as inhibitors of Phytophthora infestans. This report identifies the potential of this natural product derivative as a novel fungicide. The subject is actual and contribute new knowledge to plant protection against Phytophthora infestans. Major revision is necessary before considering manuscript for publication.
Keywords should reflected the scientific content of the work and should not be repetition of the title words (Phytophthora infestans, ellipticine). Please find such words which are not in the title, this way search engines of the web will find your manuscript with higher probability.
The aim of the study must be clearly stated.
Are all figures needed in the Introduction section?
There is not enough detail about the experimental design. It is not clear what substances are tested, the number of replication.
In the Results section, unclear whether this is a results of this paper or previous work. Many references are cited.
Discussion is too general. Why for this section were used only data from day five? Why the results of this study were not compared with previous research? No references are cited.
Line 372” Figure 11 or Figure 15?
This section should be shortened. In Conclusion should not be used "In summary" (Line 780).
Author Response
The authors would like to thank the reviewer for their comments on our manuscript and have endeavoured to address each one in specific in the revised manuscript. The response to each specific comment is outlined below in red italics:
Reviewer 1
The manuscript describes a laboratory test experiment on synthesis and evaluation of novel ellipticines and derivatives as inhibitors of Phytophthora infestans. This report identifies the potential of this natural product derivative as a novel fungicide. The subject is actual and contribute new knowledge to plant protection against Phytophthora infestans. Major revision is necessary before considering manuscript for publication.
Keywords should reflected the scientific content of the work and should not be repetition of the title words (Phytophthora infestans, ellipticine). Please find such words which are not in the title, this way search engines of the web will find your manuscript with higher probability. Thank you for this suggestion, the keywords have been amended as follows: Ellipticinium salts; Isoellipticines; Mycelial growth; Oomycete; Zoosporogenesis.
The aim of the study must be clearly stated. The following text has been added: L94 The aims of this study are to synthesise 2- and 6-substituted 9-formylellipticine derivatives by expansion with alkyl substituents and evaluate these new compounds in assays of P. infestans mycelial growth and zoosporogenesis (the production of the motile zoospore stage).
Are all figures needed in the Introduction section? Thank you for this suggestion – while this is an excellent point, this manuscript contains synthetic chemistry, the biology of plant and mammalian cells and the development of new fungicidal agents against P. Infestans and for this reason must be accessibile by multiple reader types. There are only four figures in the introduction and these cover the state of the art (Figures 1 and 2) and the discovery of ellipticines against P. infestans (Figure 3 and 4). The final figure illustrates the synthetic aims of this work to increase accessibility of the work. If insisted on by the reviewer/editorial team we will happily remove some figures but would prefer not to.
There is not enough detail about the experimental design. It is not clear what substances are tested, the number of replication. The scope of experimental design from a synthetic perspective is identified in Figure 5 and from a bioassay perspective in section 2.2 and in the footnote of each Figure. The following text has been added to clarify fully:
L153 Nineteen synthetic ellipticine derivatives were assessed in a parallel approach, via the determination of inhibitory effects on the mycelial growth, zoosporogenesis and zoospore motility in Phytophthora infestans and cytotoxicity against a mammalian cell-line, human embryonic kidney-293T (HEK-293T) and all data reported is the mean of three individual experiments
In the Results section, unclear whether this is a results of this paper or previous work. Many references are cited. Citations have been used for two purposes in the results section: in the first case there is a single citation to our paper submitted to Pathogens on the discovery of ellipticines and cinnamaldehyde derivatives as new lead against P. infestans and the rationale for this work [21]. Any other citation in the Results section is to reference the use of an experimental method used for the first time in this context which is the convention in both synthetic chemistry and bioassay. Results for compounds 15-19 are included from [21] in order that a complete comparison of the ellipticine library can be undertaken.
Discussion is too general. Why for this section were used only data from day five? Why the results of this study were not compared with previous research? No references are cited. Thank you for this comment – we have addressed the general discussion by incorporation of the following:
Reference to the data from previous studies has been incorporated [21, 33, 34].
Data from day 5 was used as the primary source of comparison as this represents the initial and most consistent P. infestans growth period which enables true comparison of compound growth effects. Data from day 9 and 13 and zoosporogenesis have been added to the discussion to remedy this as follows:
L347 Remarkably, the effects of the ellipticinium salts on P. infestans growth from one 25 mM dose last through 9 and beyond 13 days (and even as far as 35 days – data not included). This is particularly true of compounds 20 and 21 which still register 0% growth with limited effects also seen for 22, 26, 27, 31 and 32 and all other compounds registering no remaining inhibitory effects which serves to reinforce the importance of short chain alkyl substituents for inhibition. The effects on zoosporogenesis across the panel are also highly instructive with 20-22 and 31-32 effectively abolishing the production of motile zoospores.
Line 372” Figure 11 or Figure 15? Not changed – the reference to Figure 11 is designed to focus on the potency difference between cinnamaldehyde and compound 21. This highlights the significant impact of our discovery.
This section should be shortened. In Conclusion should not be used "In summary" (Line 780). The Conclusion section is 20 lines long and covers the essential findings of our work – I am not sure they can be shortened further without loss of some impact but would be happy to do so if the reviewer/editorial team suggest so. This text “In summary” has been deleted as directed.
Thank you very much for your comments
Reviewer 2 Report
It would have been interesting to have a bioessay with inoculated potato plants but I understand this could be part of a further paper.
Author Response
The authors would like to thank the reviewer for their comments on our manuscript and have endeavoured to address each one in specific in the revised manuscript. The response to each specific comment is outlined below in red italics:
It would have been interesting to have a bioessay with inoculated potato plants but I understand this could be part of a further paper.
Thank you very much for your comments. The excellent suggestion to extend the study with a field trial is a key objective of our current work and we hope to publish this at a future date.
Round 2
Reviewer 1 Report
Accept in present form.